

# eGAC3D: enhancing depth adaptive convolution and depth estimation for monocular 3D object pose detection

Duc Tuan Ngo[1,2], Minh-Quan Viet Bui[1,2], Duc Dung Nguyen[1,2] and Hoang-Anh Pham[1,2]

[1] Ho Chi Minh City University of Technology (HCMUT), Ho Chi Minh, Vietnam
[2] Vietnam National University Ho Chi Minh City (VNU-HCM), Ho Chi Minh, Vietnam

## ABSTRACT

Many alternative approaches for 3D object detection using a singular camera have been studied instead of leveraging high-precision 3D LiDAR sensors incurring a prohibitive cost. Recently, we proposed a novel approach for 3D object detection by employing a ground plane model that utilizes geometric constraints named GAC3D to improve the results of the deep-based detector. GAC3D adopts an adaptive depth convolution to replace the traditional 2D convolution to deal with the divergent context of the image's feature, leading to a significant improvement in both training convergence and testing accuracy on the KITTI 3D object detection benchmark. This article presents an alternative architecture named eGAC3D that adopts a revised depth adaptive convolution with variant guidance to improve detection accuracy. Additionally, eGAC3D utilizes the pixel adaptive convolution to leverage the depth map to guide our model for detection heads instead of using an external depth estimator like other methods leading to a significant reduction of time inference. The experimental results on the KITTI benchmark show that our eGAC3D outperforms not only our previous GAC3D but also many existing monocular methods in terms of accuracy and inference time. Moreover, we deployed and optimized the proposed eGAC3D framework on an embedded platform with a low-cost GPU. To the best of the authors' knowledge, we are the first to develop a monocular 3D detection framework on embedded devices. The experimental results on Jetson Xavier NX demonstrate that our proposed method can achieve nearly real-time performance with appropriate accuracy even with the modest hardware resource.

# INTRODUCTION

Various methods based on human visual perception (*Ren et al., 2015*; *He et al., 2017*; *Redmon & Farhadi, 2018*; *Zhou, Wang & Krähenbühl, 2019*; *Tan, Pang & Le, 2020*) have been proposed to solve 2D object detection tasks and achieve exceptional performance by utilizing the power of the deep neural network in computer vision (*Krizhevsky, Sutskever & Hinton, 2017*; *He et al., 2016*; *Brock et al., 2021*). The requirement for scene understanding, which includes detailed 3D poses, identities, and scene context, is still high in some domains, such as autonomous driving or infrastructure-less robot navigation. As a result,

Corresponding author
Hoang-Anh Pham,
anhpham@hcmut.edu.vn

3D object pose detection has recently received interest from academics and industries, particularly in autonomous navigation applications.

Numerous LiDAR sensors are being used worldwide to create an accurate depth map of the environment for 3D object detection purposes. LiDAR sensors are popular because they produce a dependable 3D point cloud that is gathered using laser technology. Although such LiDAR-based systems (*Shi, Wang & Li, 2019*; *Lang et al., 2019*; *Shi et al., 2020*; *He et al., 2020*) have produced promising results, they also have significant drawbacks, including expensive, difficult to mount on automotive vehicles, and sparse and unstructured data. An alternative approach based on a single RGB camera is considered as a result of this situation. Compared to LiDAR sensors, it is far more affordable, adaptable, and universally available on modern automobiles. Due to the lack of depth information, camera-based methods have worse performance in image-based 3D object detection than LiDAR-based approaches. Though stereo cameras are available to produce high resolution, we must calibrate the cameras with a high degree of precision. However, the quality of the depth map obtained falls well short of the grade required for autonomous driving systems, particularly in outdoor environments. The use of a single camera to estimate depth is a considerable choice because humans can perceive depth from 2D images as well as 3D images. The accuracy of monocular depth estimation, on the other hand, is not as good as that of ToF cameras such as LiDAR. This makes 3D object detection on a single camera more complicated, and it has attracted a great deal of attention from the scientific community.

The monocular 3D object pose detection can be formulated as follows: given a single RGB image $I \in R^{H \times W \times 3}$, where $H$ and $W$ are the image sizes, and the camera parameters such as focal length, principle center, relative translation and rotation, the target is for each object of interest (car, cyclist, pedestrian), classify the category $C$ and its 3D bounding box $B$ which is parameterized with nine variables: 3D center location $(c_x, c_y, c_z)$, dimension $(h, w, l)$ and orientation $(\theta_{roll}, \theta_{pitch}, \theta_{yaw})$. In 3D detection task, the location and dimension are measured by the absolute value (in meters), compared with the relative value (in pixels) of the 2D object detection task. Typically, the roll and pitch angles are set to zero by default. Figure 1 shows an example to describe the difference between 2D and 3D detection.

Recently, we have proposed an improved monocular 3D object detection named GAC3D (*Bui et al., 2021*), which leverages the assumption that most objects of interest stand on a ground plane. This hint serves as a solid geometric prior to estimating the object's distance *via* 2D-3D re-projection. Moreover, we introduced the depth adaptive-convolution that enhances the extracted features from the backbone network to consolidate the performance of detection heads. The depth adaptive convolution takes an extra input, a depth estimation map, to re-weight the parameters of convolution kernels. The experimental results on the KITTI 3D Object Detection benchmark have shown that GAC3D achieves better detection results than many existing methods at that time. However, as GAC3D still depends on an external depth estimator to generate the depth map, its architecture is split into two sub-models (one for depth estimation and one for 3D

PeerJ Computer Science

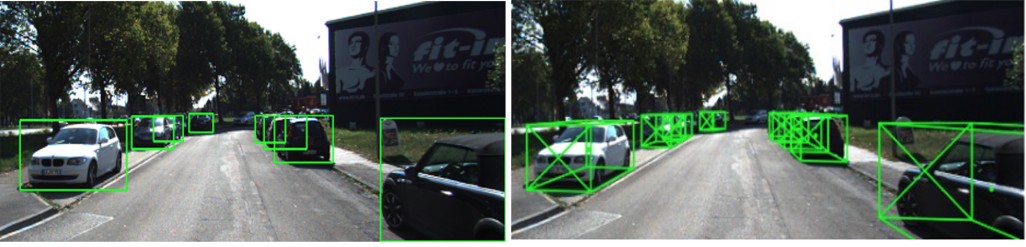

**(A)** 2D detection example        **(B)** 3D detection example

**Figure 1** **2D detection is to determine the bounding box in 2-dimension space (image) including: x, y, width, and height (A).** Meanwhile, 3D detection is to determine the bounding box in 3-dimension space (real-world space) including: x, y, z, length, width, height, and orientation (B).

object detection) then it takes a long inference time and it is not fully end-to-end optimized.

Another crucial metric to assess the performance of an embodied perception task is the reaction time (*Li, Wang & Ramanan, 2020*). Practical applications, such as self-driving vehicles, or augmented reality and virtual reality (AR/VR), may necessitate reaction times that are on par with human capabilities. In such requirements, low-latency algorithms are necessary to guarantee the safe operation, or to provide a truly immersive experience for users. However, earlier works on 3D object detection have not particularly focused on algorithmic latency, making them challenging for applying in real-world applications. Some recent approaches have provided feasible solutions which can balance both accuracy and efficiency. These methods can achieve ideal throughput for the real-time requirement with sufficient computational resources. Nonetheless, robotics and autonomous driving systems require multiple perception and navigation tasks on limited computing hardware resources.

This article presents an alternative architecture inheriting the ideas from our previous work GAC3D. We name this method eGAC3D (enhanced GAC3D) which has two main contributions.

- The proposed eGAC3D is a multi-task learning model which simultaneously estimates the depth prior and performs monocular 3D object detection. Accordingly, we can reduce the framework's complexity, make the model end-to-end optimized and significantly reduce the inference time.
- We present the detailed deployment process of the proposed eGAC3D on a low-priced and modest computational hardware platform such as the Jetson Xavier NX embedded board. The experimental results show that our proposed eGAC3D can achieve real-time performance while maintaining accuracy in the mainframe computer.

The remainder of this article is organized in the following manner. The next section presents a brief survey of the state-of-the-art methods for monocular 3D object detection in the last 3 years as our related work. Then, we describe our eGAC3D architecture compared with previous work GAC3D. We demonstrate the implementation and

Ngo et al. (2022), *PeerJ Comput. Sci.*, DOI 10.7717/peerj-cs.1144

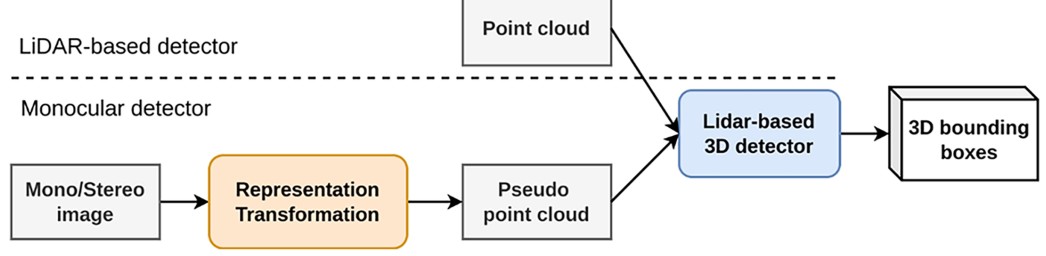

**(A)** Data-lifting approaches

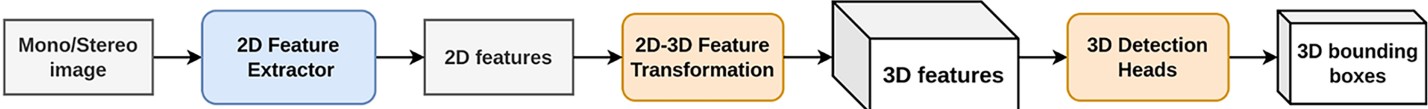

**(B)** Feature-lifting approaches

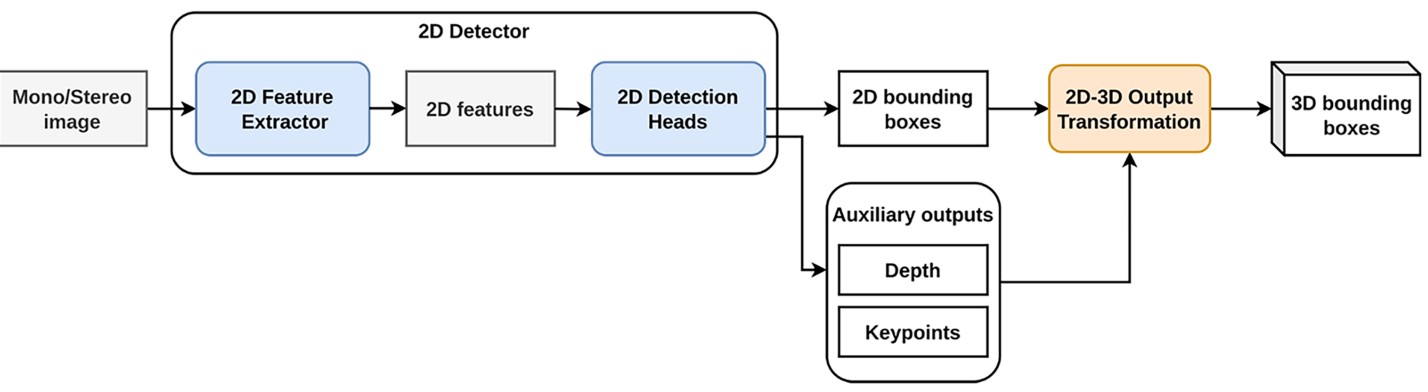

**(C)** Result-lifting approaches

**Figure 2 (A–C) Three common approaches of monocular 3D object detection.**

evaluation of eGAC3D performance on KITTI benchmark compared to the most recent methods. The last section provides the final concluding remarks and future works.

## RELATED WORK

We characterize state-of-the-art (SOTA) approaches to monocular 3D object detection in the last 3 years into three main categories, including Data-lifting, Feature-lifting, and Result-lifting approaches based on the position of the 2D-3D transformation in the pipeline as illustrated in Fig. 2.

## Data-lifting approaches

This line of works researches how to transform the monocular image to mimic the 3D point cloud input of LiDAR-based detection methods. Common approaches usually generate pseudo point clouds from the pixel-wise depth estimation of RGB images and converting perspectives images to birds-eye-view (BEV) images. Then, a 3D object detector is applied to perform the detection task from the pseudo point clouds like LiDAR-based approaches. Representation transformation-based approaches significantly improve the performance of monocular object detection results. However, these approaches require many complex computing stages and take long inference time. *Wang et al. (2019)* proposed a method using state-of-the-art depth estimation to generate pseudo point cloud before directly applying LiDAR-based 3D object detection algorithms. By leveraging the estimated depth map $D$ as an additional signal along with the RGB image, the 3D location $(x, y, z)$ of each pixel $(u, v)$ could be derived as follows:

$$x = \frac{(u - c_U)z}{f_U} \tag{1}$$

$$y = \frac{(v - c_V)z}{f_V} \tag{2}$$

$$z = D(u, v) \tag{3}$$

where $(c_U, c_V)$ is the pixel location corresponding to the camera center and $f_U, f_V$ are the vertical and horizontal focal length.

*You et al. (2020)* offered Pseudo-LiDAR++ as an enhancement to their previous work (*Wang et al., 2019*). They modified the stereo network architecture and loss function in order to make them more aligned with a precise depth estimation of distant objects. Then, they utilized highly sparse, but less expensive LiDAR sensors for debiasing depth estimation and improving the correctness of point cloud representation.

*Wang et al. (2020)* introduced a noteworthy method called ForeSeE based on pseudo-LiDAR approaches. According to the authors' finding, foreground and background have different depth distributions so their depth may be separately determined by using distinct optimization objectives and decoders. This work significantly improves the depth estimation performance on foreground objects.

## Feature-lifting approaches

*Roddick, Kendall & Cipolla (2019)* proposed the orthographic feature transform (OFT), which constructs an orthographic birds-eye-view from the corresponding monocular pictorial cues. Specifically, each voxel of the 3D feature map is the average pooling of the image-based features over the projected voxel area. Then, in order to preserve the computational efficiency, they accumulate the 3D voxel along the vertical axis to create the birds-eye-view orthographic features. Finally, the orthographic features are processed by the topdown convolutional network to estimate the objects' locations and poses directly.

Recently, *Reading et al. (2021)* have introduced a new method following the representation transformation approach named CaDDN. The key idea is to project the perspective image to BEV representation and then perform object detection. In this work,

rich contextual feature information is projected to the appropriate depth interval in 3D space based on a predicted categorical depth distribution for each pixel. However, instead of separating depth estimation from 3D detection during the training phase like previous methods, the authors performed end-to-end training to focus on generating an interpretable depth representation for monocular 3D object detection.

### Result-lifting approaches

Result-lifting methods leverage the existing 2D object detection frameworks to directly estimate the object's 2D location with its corresponding depth, dimension, and orientation from 2D features. *Liu, Wu & T'oth (2020)* employed the CenterNet (*Zhou, Wang & Krähenbühl, 2019*) architecture to predict object's projected 3D center with its corresponding depth, dimension and orientation. Then, they reconstructed the object's 3D location *via* 2D-3D projection. The authors also introduced a multi-step disentangling loss to enhance the optimization robustness. *Zhou et al. (2021)* improved the above design by incorporating object instance segmentation to collect the instance-aware 2D features extracted from the backbone. Instead of directly predicting the projected 3D keypoints from 2D features, *Li et al. (2020)* estimated the offset of the keypoints to the 2D object's center. They formalized a post-optimization process from the object's keypoints, dimensions, and orientation to retrieve the final object's location in 3D space. To perform an end-to-end training pipeline, *Li & Zhao (2021)* transformed the post-optimization step to a differentiable loss and integrated it into the training phase. *Ma et al. (2021)* found that far-away objects are almost impossible to localize accurately by using only monocular image and adding the samples in the training phase make the model hard to be optimized. They proposed the hard and soft codings to handle improper samples, where the hard coding eliminates all the distant objects by a fixed threshold, and the soft coding re-weights the impacts of training objects according to their ground-truth distance.

*Brazil & Liu (2019)* exploited the anchor-based object detection approach for monocular 3D detection. The authors introduced a single-stage network with the concept of a 2D-3D anchor box to predict 2D and 3D boxes simultaneously. Furthermore, they introduced depth-aware convolution to improve the spatial awareness of high-level features. *Ding et al. (2020)* also followed the 3D anchor-based design and utilized the external depth maps guidance to learn the geometric-aware deep representation for the detection network. Besides, *Liu, Yixuan & Liu (2021)* proposed another strategy to minimize the gap between 2D and 3D representation by leveraging the fact that most objects of interest should be on the ground plane. The authors accumulated the 2D center's feature with the corresponding contact point's feature to gather the useful structural information. *Kumar, Brazil & Liu (2021)* designed a differentiable non-maximum-suppression (NMS) for monocular 3D object detection and directly calculate the loss on the results after NMS step.

### DISCUSSION

Monocular 3D object detection is an ill-posed problem due to the lack of depth information. Therefore, directly regressing properties of 3D bounding boxes is such a

challenge. Numerous approaches have been continuously proposed to improve performance. The data-lifting methods adopt the existing LiDAR-based detector with different representations of image input such as the pseudo-point cloud, BEV image, and frustum. These methods heavily depend on the accuracy of the intermediate representation of monocular image and they are hard to be optimized due to the complicated pipeline process. The feature-lifting approaches apply the 2D-3D transformation in the feature space, instead of the input space. Overall, feature-lifting and data-lifting approaches consume the huge computational cost in both time and memory due to 3D convolution and these methods only work well when the input data already contain geometrical cues. Therefore, these methods are commonly applied in other tasks such as stereo 3D object detection, stereo matching or multi-view-stereo (*Chen et al., 2020*; *Guo et al., 2021*; *Gu et al., 2020*; *Yao et al., 2018*), but not in monocular 3D object detection.

Among various results-lifting approaches, several studies are to utilize virtual keypoints based on 2D bounding boxes and leverage geometric constraints on 2D-3D relationship to optimize the predictions. While other methods use monocular depth estimation to generate additional input depth maps to narrow the representation gap between 3D point clouds and 2D images. Most of the recent methods are extending either anchor-based detector or center-based detector methods that have consistently shown improved performance. Compared to the anchor-based approach, the center-based one is currently more popular due to its flexibility and computational efficiency. Our proposed GAC3D and eGAC3D are also center-based methods.

## THE PROPOSED APPROACH

As illustrated in Fig. 3, the detection network in eGAC3D architecture inherits the idea of our previous GAC3D architecture, which consists of a backbone for feature extraction followed by multiple detection heads. The detection network takes a monocular RGB image $I \in R^{H \times W \times 3}$ where $H$ and $W$ is the image height and width to produce the intermediate prediction of the object's location, dimension, orientation, and a set of projected object's keypoints. Every detection head shares a mutual backbone network. The extracted features from the backbone network are sequentially passed through a convolution layer with $3 \times 3$ filters, a ReLU activation and a $1 \times 1$ convolution layer. Each head produces a tensor having a resolution of $\left(\dfrac{H}{R}, \dfrac{W}{R}\right)$ with $C$ channels, where $R$ is the output stride and $C$ is distinct for each head.

In both eGAC3D and GAC3D, the backbone network encodes the input into a high-dimensional feature representation to perform prediction tasks. We apply the deep layer aggregation backbone by adding more skip connections to the original DLA-34 network (*Yu et al., 2018*) and replacing the standard convolution layers in the upsampling nodes with deformable convolution layers proposed in (*Dai et al., 2017*). Finally, the backbone generates the feature maps of size $\dfrac{H}{4} \times \dfrac{W}{4} \times 512$.

Typically, vehicles obstruct others in traffic scenes. We observe that irrelevant features, such as those belonging to adjacent or background objects, should not influence the detection result of a specific object. Instead of using instance segmentation as a guide for

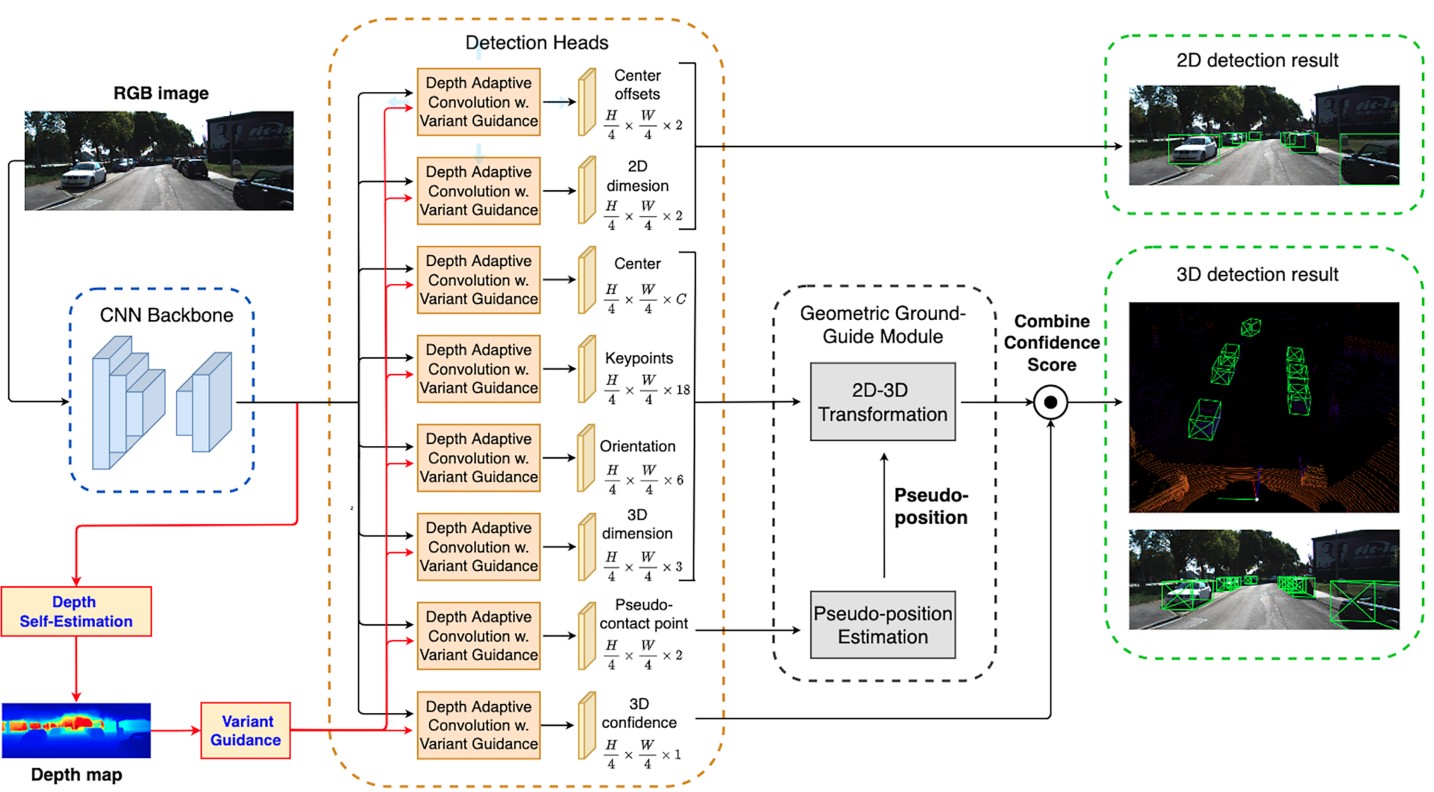

**Figure 3** The proposed eGAC3D's overview architecture is a multi-task learning model which simultaneously estimates the depth prior by the Depth Self-Estimation and performs monocular 3D object detection. Additionally, eGAC3D adopts a revised depth adaptive convolution with Variant Guidance to predict the object center, pseudo-contact point offset, keypoints offset, observation angle, and dimension.

the detection network, we use the 3D surface of the object taken from the depth map to select valuable features from pixels inside the object that should contribute more to 3D detection. By probing the depth map, we can always determine the discontinuity between objects. To deal with the divergent context of the image's feature, we design a Depth Adaptive Convolution layer to replace the traditional 2D convolution in detection heads. Figure 4 shows an example to demonstrate the correlation between the depth map and the instance segmentation of objects. If we look closer at the discontinuity in the depth map, we can figure the boundary of each object.

## Depth adaptive convolution with variant guidance

The depth adaptive convolution layer proposed in our previous work GAC3D (*Bui et al., 2021*) injects information from depth predictions using pixel-adaptive convolution from *Su et al. (2019)*. We design the detection head with a sequence of a depth adaptive convolution layer with $3 \times 3$ filters, a ReLU activation, and a $1 \times 1$ standard convolution layer. The depth maps generated from pre-trained models are scaled down by the bilinear interpolation with the factor of $\frac{1}{4}$. The scaled depth estimations are then directly used as the guidance for depth adaptive convolution. The kernel's parameters, such as padding values,

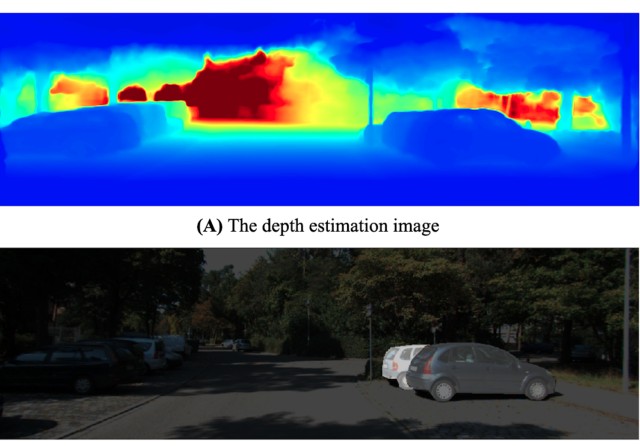

**(A)** The depth estimation image

**(B)** The instance segmentation of objects in the image

**Figure 4** **The relative correlation between the depth surface (A) and the objects in the image (B).**

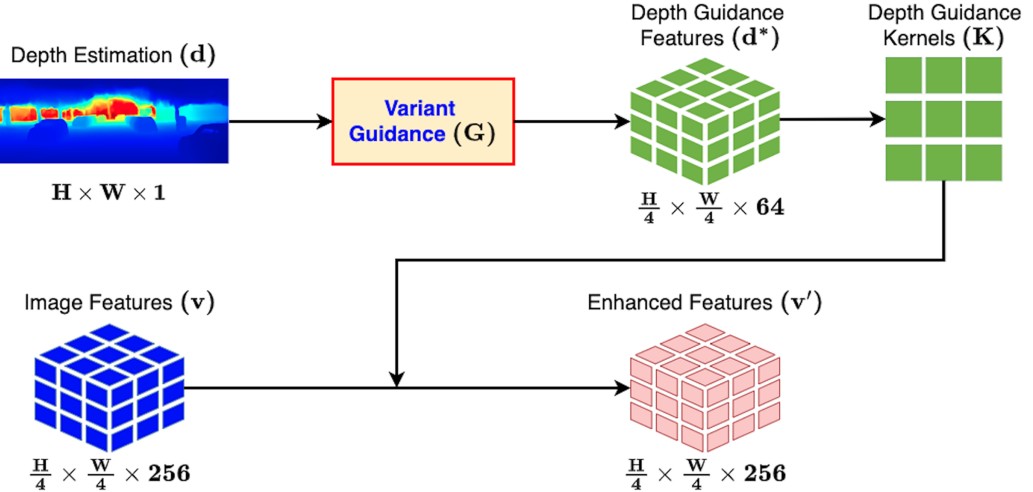

**Figure 5** **Depth adaptive convolution detection head with variant guidance in eGAC3D.** The variant guidance function transforms the pixel-wised depth estimation before using it in the depth adaptive convolution operation.

strides remain unchanged. Hence, each depth adaptive detection head has the exact size of input and output.

On account of the 2D-3D projection properties, the monocular pixels of far distance objects would critically scale in terms of depth value. Hence, treating every pixel-wise depth with the exact guidance is not sufficient enough. We improve the depth adaptive convolution by introducing a learnable guiding kernel called **variant guidance** in eGAC3D. This innovative guidance allows the network to learn the scale sensitivity of monocular depth estimation implicitly and smoothly adapt the guidance weights basing on both the local object features and global structure information. Figure 5 illustrates the proposed Depth Adaptive Convolution with Variant Guidance in eGAC3D while its formulation is adjusted by (4).

$$\begin{cases} \mathbf{d}^* = G(d) \\ \mathbf{v_i}' = \sum_{j \in \Omega(i)} K(\mathbf{d_i^*}, \mathbf{d_j^*}) \mathbf{W}[\mathbf{p_i} - \mathbf{p_j}] \mathbf{v_j} + \mathbf{b} \end{cases} \quad (4)$$

where $G$ is the variant guidance function to compute a variant vector $\mathbf{d}^*$ of the depth estimation value $d$. We implement $G$ by applying a sequence of standard convolution layers and ReLU activations. $G$ comprises two consecutive block of $3 \times 3$ convolution with the stride of two followed by ReLU activation and a $1 \times 1$ convolution layer.

## Depth self-estimation

The investigation by *Wang et al. (2020)* showed that foreground depth and background depth of traffic scenes show different data distributions. As our foremost task is 3D object detection, the precision of the foreground depth plays a crucial role in object localization. However, pre-trained depth estimators typically penalize errors across all pixels equally. Hence, using pre-trained depth estimators would lead to over-emphasizing non-object pixels problems and consequently decrease the object detection performance. Moreover, employing a separate depth estimator requires redundant computation cost and critically increases the framework's inference time. To overcome the above issues, we design an end-to-end training strategy for depth guided detection network. In eGAC3D, we create a new head in the detection network to estimate depth values itself as shown in Fig. 3 instead of leveraging an external depth estimator trained on a generic depth estimation image *corpus* as our previous work GAC3D.

The depth branch processes the feature maps with a $3 \times 3$ convolution with ReLU activation followed a $1 \times 1$ convolution with Sigmoid activation to produce a low resolution of depth estimation $\in R^{\frac{H}{4} \times \frac{W}{4} \times 1}$. Then, it is interpolated with the nearest neighbor mode to produce the final depth estimation with full resolution $\in H \times W \times 1$.

## Geometric ground-guide module

In eGAC3D, we adopt this Geometric Ground-Guide Module (GGGM) proposed in GAC3D to leverage the output from the detection network, including *orientation*, *dimension*, *keypoints' position*, and *pseudo-contact point's position* for reconstructing the 3D bounding box. GGGM begins by estimating a pseudo-position for each detected object. The detection network outputs including the 2D center, dimension, orientation, nine predicted keypoints, and the pseudo-position are then used to generate a system of geometric equations to compute the position of the 3D bounding box center. Figure 6 depicts an example of how GGGM works, while the details of its computation were presented in GAC3D.

## Confidence score

The confidence score directly affects the evaluation of detection results. In the 2D object detection scope, the confidence score denotes the probability that a bounding box containing an object. A high-confidence object also indicates that it could be easily discriminated from the 2D image. However, because of the occlusion, truncation, and distance variance of objects in 3D space, objects assigned with a high 2D confidence score

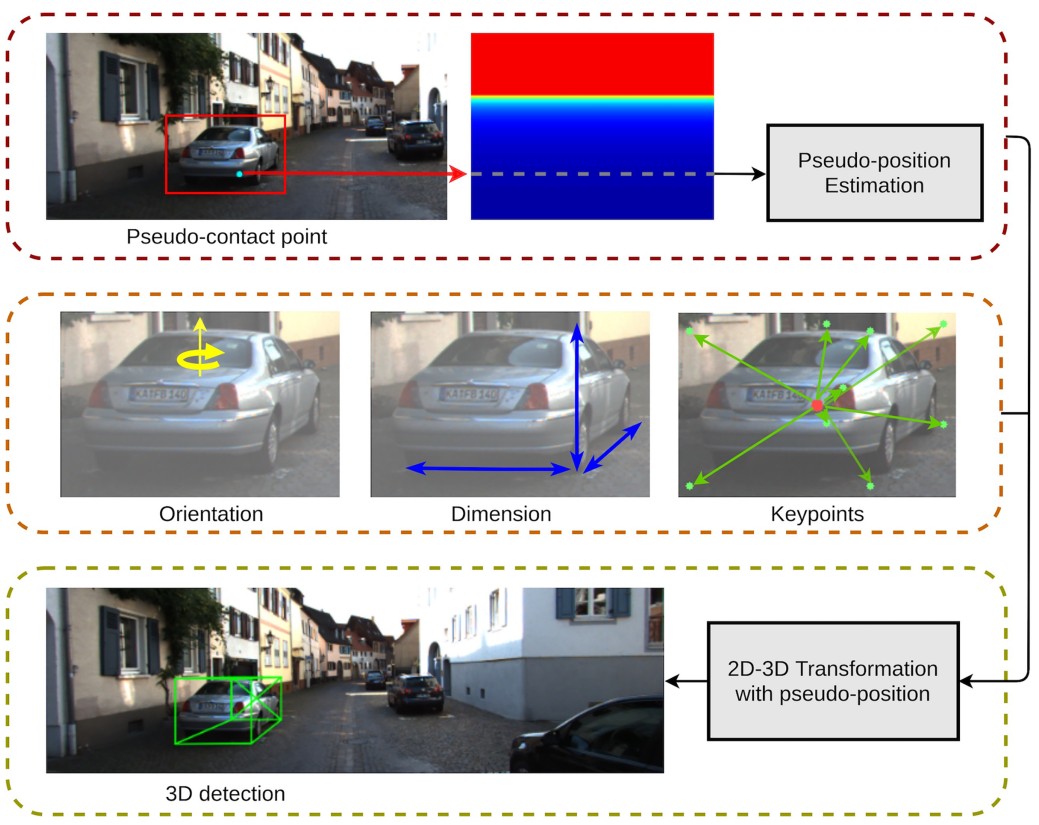

**Figure 6** An illustration of the geometric ground-guide module.

could not be located easily in 3D space. Thus, it is inadequate to naively employ original 2D object detection confidence as a 3D confidence score. Moreover, due to the lack of label information, we could not train our model to directly regress the confidence of a 3D detection. To resolve the current problem, we follow the object confidence decomposition proposed in *Li & Zhao (2021)*. This 3D object confidence score estimation combines both 2D and 3D perspective detection mechanism by computing a prior distribution as follows:

$$con\ f = con\ f_{2D} \times con\ f_{3D} \tag{5}$$

where *con f* is the object's confidence score, *con $f_{2D}$* is the 2D confidence score and extracted from the output heatmap of center head. The 3D confidence head predicts the 3D bounding box confidence $Con\ f_{3D} \in R^{\frac{H}{4} \times \frac{W}{4} \times 1}$ by the IoU between estimation and groundtruth. Figure 7 gives an example of the correlation between 3D confidence scores estimated from our detection network and the quality of the predicted bounding boxes.

# IMPLEMENTATION AND EVALUATION

## Implementation

Similar to our previous work GAC3D, we implement and train eGAC3D on a machine equipped with an Intel Core i7-9700K CPU, a single RTX-2080ti GPU 12 GB, and 32 GB main memory. For DLA-34 backbone network, we first load the ImageNet pre-trained

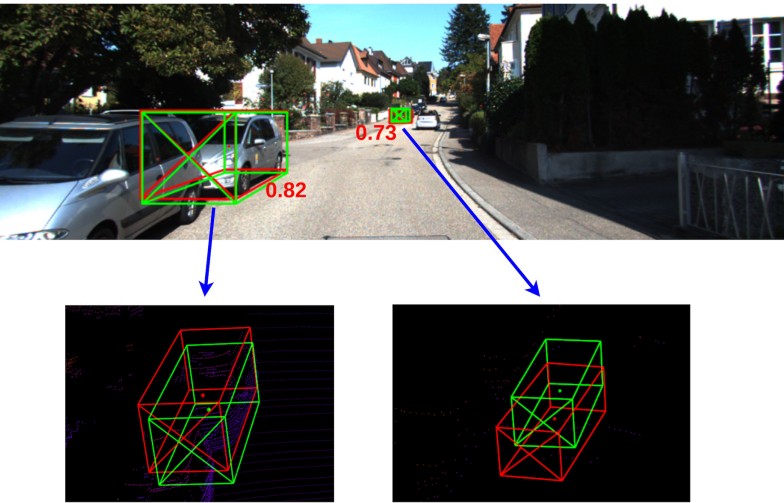

**Figure 7** **Illustration of 3D confidence outputs.** Green: prediction, red: ground truth. The further car with less accurate 3D pose estimation yields a lower 3D confidence score.

weights which are provided in the Pytorch framework by default. The other weights of our framework are trained from scratch. Our DLA-34 backbone networks contain 16.85 million parameters.

## Evaluation

We conduct experiments to evaluate our eGAC3D performance on the official KITTI object detection benchmark (*Geiger, Lenz & Urtasun, 2012*) that consists of four different benchmarks, including 2D detection, orientation similarity, 3D detection, and BEV detection on three classes (*Car*, *Pedestrian*, and *Cyclist*). These benchmarks are separated into three difficulty levels *Easy*, *Moderate*, and *Hard* according to the bounding box's height, occlusion and truncation level (*Geiger, Lenz & Urtasun, 2012*). All benchmarks are ranked by the precision-recall curves given the IoU threshold or rotation threshold. From October 2019, following *Chen et al. (2015)*, the evaluation metrics are changed from 11-point Interpolated Average Precision (AP) metric $AP|_{R11}$ to 40 recall positions-based metric $AP|_{R40}$. For evaluating 3D detection performance, we mainly focus on the 3D detection, and BEV detection metrics, especially for the *Car* category. For *Car* objects, it requires an 3D bounding box overlap of 70% and for *Cyclist* and *Pedestrian* is 50%. Table 1 shows the detailed evaluation results of eGAC3D performance. We also highlight the 3D detection and BEV detection on the *Car* category that is the main consideration in our study.

### *Comparative results*

We conduct a comprehensive performance comparison between our eGAC3D with the most recent methods, including our previous work GAC3D based on the KITTI 3D object detection benchmark. Table 2 shows comparative results of 13 recent methods, which are ordered by the $AP|_{R40}^{3D}$ of the *Moderate* difficulty level of the *Car* category. It can be observed that our eGAC3D achieves state-of-the-art results on KITTI benchmark for *Car*

**Table 1 KITTI object detection benchmark on *test* set of the proposed eGAC3D.** The top 2 highest results are bold highlighted.

| Benchmark | $AP\|_{R40}$ | | |
|---|---|---|---|
| | **Easy** | **Mod** | **Hard** |
| Car (2D det.) | 92.12 | 79.40 | 69.52 |
| Car (orientation) | 92.09 | 79.33 | 69.45 |
| **Car (3D det.)** | **19.73** | **13.97** | **11.82** |
| **Car (BEV det.)** | **27.41** | **19.03** | **16.46** |
| Cyclist (2D det.) | 39.34 | 24.46 | 19.59 |
| Cyclist (orientation) | 34.09 | 20.74 | 16.83 |
| Cyclist (3D det.) | 7.36 | 4.42 | 3.34 |
| Cyclist (BEV det.) | 8.31 | 4.99 | 4.08 |
| Pedestrian (2D det.) | 29.93 | 19.96 | 17.46 |
| Pedestrian (orientation) | 29.09 | 19.40 | 16.95 |
| Pedestrian (3D det.) | 8.79 | 5.95 | 5.72 |
| Pedestrian (BEV det.) | 10.33 | 6.15 | 5.99 |

**Table 2 Comparative results on the KITTI 3D object detection *test* set of the *Car* category.** The top 2 highest results are bold highlighted.

| Method | Backbone | $AP\|_{R40}^{3D}$ (IoU = 0.7) | | | $AP\|_{R40}^{BEV}$ (IoU = 0.7) | | | Runtime (ms) |
|---|---|---|---|---|---|---|---|---|
| | | **Easy** | **Mod** | **Hard** | **Easy** | **Mod** | **Hard** | |
| *Brazil & Liu (2019)* | DenseNet-121 | 14.76 | 9.71 | 7.42 | 21.02 | 13.67 | 10.23 | 160 |
| *Liu, Wu & T'oth (2020)* | DLA-34 | 14.03 | 9.76 | 7.84 | 20.83 | 14.49 | 12.75 | 30 |
| *Li et al. (2020)* | DLA-34 | 14.41 | 10.34 | 8.77 | 19.17 | 14.20 | 11.99 | 55 |
| *Li & Zhao (2021)* | DLA-34 | 16.73 | 11.45 | 9.92 | 23.44 | 16.20 | 14.47 | 40 |
| *Ding et al. (2020)* | ResNet-50 | 16.65 | 11.72 | 9.51 | 22.51 | 16.02 | 12.55 | 200 |
| **GAC3D** (*Bui et al., 2021*) | DLA-34 | 17.75 | 12.00 | 9.15 | 25.80 | 16.93 | 12.50 | 250 |
| *Zhou et al. (2021)* | DLA-34 | 17.81 | 12.01 | 10.61 | 25.88 | 17.88 | 15.35 | 34 |
| *Ma et al. (2021)* | DLA-34 | 17.23 | 12.26 | 10.29 | 24.79 | 18.89 | 16.00 | 40 |
| *Chen et al. (2021)* | ResNet-101 | 19.65 | 12.30 | 10.58 | 27.94 | 17.34 | 15.24 | 70 |
| *Kumar, Brazil & Liu (2021)* | ResNet-101 | 18.10 | 12.32 | 9.65 | 26.19 | 18.27 | 14.05 | 120 |
| *Liu, Yixuan & Liu (2021)* | ResNet-101 | **21.65** | 13.25 | 9.91 | **29.81** | 17.98 | 13.08 | 50 |
| *Reading et al. (2021)* | ResNet-101 | 19.17 | 13.41 | **11.46** | 27.94 | 18.91 | **17.19** | 630 |
| **eGAC3D** | DLA-34 | 19.73 | **13.97** | **11.82** | 27.41 | **19.03** | 16.46 | **38** |

category in comparison to contemporary monocular 3D object detection frameworks. Comparing with the second best competitor, we achieve 19.73 (↑ 0.56) for *Easy*, 13.97 (↑ 0.56) for *Moderate* and 11.82 (↑ 0.36) for *Hard* while our method run much faster than many other competitors.

As summarised in Table 3, it is notable that our proposed approach also outperforms all existing monocular approaches in the *Cyclist* detection task. Additionally, eGAC3D significantly improves the performance compared to our previous GAC3D in all metrics.

**Table 3 Comparative results of the *Pedestrian* and *Cyclist* on the KITTI *test* set.** We use bold to indicate the highest result.

| Method | $\text{AP}\vert_{R40}^{3D}$ (IoU = 0.5) | | | | | | Runtime (ms) |
|---|---|---|---|---|---|---|---|
| | Cyclist | | | Pedestrian | | | |
| | Easy | Mod | Hard | Easy | Mod | Hard | |
| *Brazil & Liu (2019)* | 0.94 | 0.65 | 0.47 | 4.92 | 3.48 | 2.94 | 160 |
| *Ding et al. (2020)* | 2.45 | 1.67 | 1.36 | 4.55 | 3.42 | 2.83 | 200 |
| *Chen et al. (2021)* | 1.01 | 0.61 | 0.48 | 10.88 | 6.78 | 5.83 | 70 |
| *Reading et al. (2021)* | 7.00 | 3.41 | 3.30 | **12.87** | **8.14** | **6.76** | 630 |
| **eGAC3D** | **7.36** | **4.42** | **3.34** | 8.79 | 5.95 | 5.72 | **38** |

**Table 4 Evaluation on accumulated improvement of our proposed methods on the KITTI val set.** "P." denotes Pseudo-position, "DA." denotes Depth Adaptive Convolution, "VDA." denotes Depth Adaptive Convolution with Variant Guidance, "DSE." denotes Depth Self-Estimation. We use bold to indicate the highest result.

| Method | $\text{AP}\vert_{R40}^{3D}$ (IoU = 0.7) | | | $\text{AP}\vert_{R40}^{BEV}$ (IoU = 0.7) | | |
|---|---|---|---|---|---|---|
| | Easy | Mod | Hard | Easy | Mod | Hard |
| Baseline | 11.56 | 10.31 | 8.94 | 19.86 | 16.33 | 14.56 |
| +P. | 16.15 | 13.17 | 11.48 | 25.17 | 19.91 | 17.63 |
| +P. +DA. (**GAC3D**) | 17.59 | 14.79 | 13.10 | 25.01 | 20.56 | 18.20 |
| +P. +VDA. (**eGAC3D**) | **21.15** | 16.17 | **14.10** | 30.93 | **23.33** | **20.59** |
| +P. +VDA. +DSE. (**eGAC3D**) | 21.02 | **16.34** | 14.08 | **31.29** | 22.91 | 19.92 |

### Accumulated impact of our proposed methods

We also conducted experiments to measure the impact of each component on our proposed designs to investigate the efficient contribution of each proposed component on the overall performance of the monocular 3D object detection task. These experimental results shows new valuable contribution of revised modules in eGAC3D when compared to our previous work GAC3D. In these experiments, we follow the default setup of *Li & Zhao (2021)* to train the baseline model.

Table 4 expresses the detailed evaluation results of our methods on the KITTI **val** set. As proved in GAC3D, the depth adaptive convolution (+DA) and the pseudo-position refinement (+P) show significant improvements in both the 3D object detection task and birds-eye view detection task. In particular, the +P method increase the baseline model's performance by nearly 28% in terms of 3D object detection with the moderate setting. Moreover, the incorporation of both +DA and +P has a relative improvement against the baseline model by 43%. We then critically improve our model by applying the depth adaptive convolution with variant guidance (+VDA) as proposed in eGAC3D. The combination of +P and +VDA archives 16.17 (↑ 1.38) comparing with the association of +P and +DA in GAC3D. Finally, we introduce the depth self-estimation strategy (+DSE) to

**Table 5 Comparisons of different depth estimation quality for 3D object detection on the KITTI *val* set.** We use (*) to indicate using standard convolution operation. Bold is used to indicate the highest result.

| Depth estimator | $AP\|^{3D}_{R40}$ (IoU = 0.7) | | | $AP\|^{BEV}_{R40}$ (IoU = 0.7) | | |
|---|---|---|---|---|---|---|
| | Easy | Mod | Hard | Easy | Mod | Hard |
| None* | 16.15 | 13.17 | 11.48 | 25.17 | 19.91 | 17.63 |
| Dorn (*Fu et al., 2018*) | 19.00 | 14.93 | 13.65 | 27.61 | 21.75 | 19.25 |
| BTS (*Lee et al., 2020*) | 19.43 | 15.75 | 13.63 | 30.35 | 22.33 | 19.43 |
| AdaBins (*Bhat, Alhashim & Wonka, 2021*) | **21.15** | 16.17 | **14.10** | 30.93 | **23.33** | **20.59** |
| DSE in eGAC3D | 21.02 | **16.34** | 14.08 | **31.29** | 22.91 | 19.92 |

bypass the requirement of pre-trained depth models and still remain the same performance while eGAC3D can significantly shorten the runtime as shown in Table 2.

***Impact of depth self-estimation quality on adaptive convolution***

In this experiment, we analyze the performance of the depth adaptive convolution with different pre-trained depth models. To study the impact of depth estimation quality on our 3D detection results, we generate different depth maps from three recent supervised monocular depth estimation methods: DORN (*Fu et al., 2018*), BTS (*Lee et al., 2020*), AdaBins (*Bhat, Alhashim & Wonka, 2021*), and our proposed Depth Self-Estimation. The generated depth maps are then applied as the guidances for the depth adaptive convolution layers. Finally, we evaluate the results on KITTI *val* set. As shown in Table 5, it is not surprising that the version using AdaBins depth estimator proposed by *Bhat, Alhashim & Wonka (2021)*, the current state of the art for monocular depth estimation, obtains higher performance than other pre-trained depth models. Notably, our depth estimation strategy provides the similar accuracy compared to using the AdaBins depth estimator, while our strategy uses a very simple network architecture for depth estimation. This justifies our claims that our end-to-end training strategy can provide a depth estimator which is well-optimized for the 3D detection task.

# DEPLOYMENT ON AN EMBEDDED SYSTEM PLATFORM

## Deployment process

Figure 8 illustrates the deployment cycle of our eGAC3D on an embedded platform, Jetson Xavier NX. As above presented, the model was trained and tested on a mainframe computer with GPU supported. When the model was properly tested and achieve appropriate results, we bring the model weight to the Jetson board by using ONNX parser to convert model's weights from Pytorch format (.pth) to ONNX format (.onnx).

With TensorRT, models can be imported from frameworks like TensorFlow and PyTorch *via* the ONNX format (.onnx) to the Jetson platform. They may also be created from scratch by instantiating individual layers and setting parameters and weights directly. In the proposed eGAC3D, we adopt a Deformable Convolution Network (DCN) in Detection Heads that is not natively supported in TensorRT. Therefore, we need to perform the model optimization to convert the model to TensorRT before building the

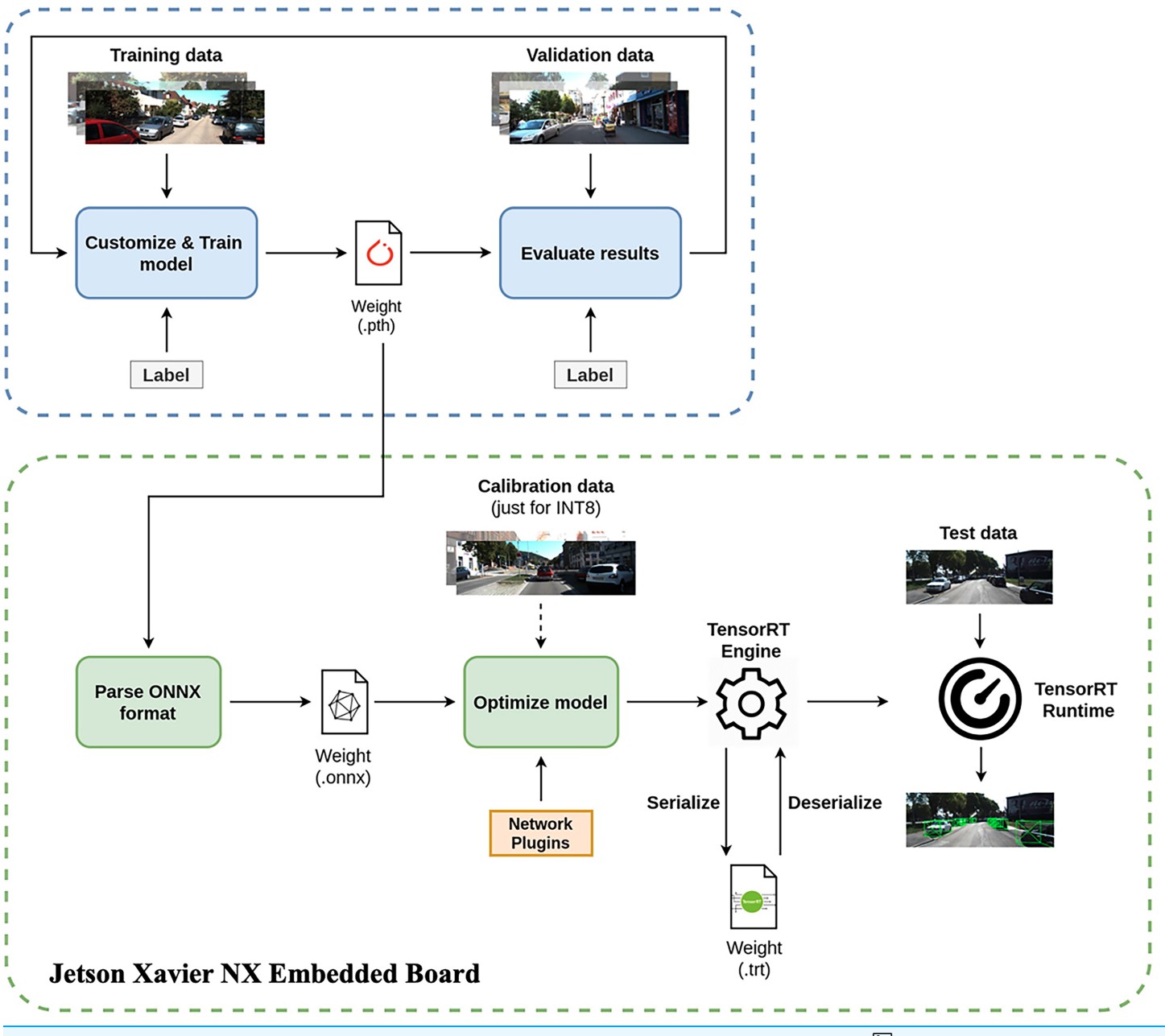

**Figure 8** Deployment process of eGAC3D on the Jetson embedded board.

TensorRT Engine. Since TensorRT already provides the interface class *IPluginV2Ext* for users to extend and implement the custom layer, we create a network plugin to implement our custom layer and add it to the engine in build time.

Regarding data calibration, TensorRT supports three types of parameter precision including 32-bit floating point (FP32), 16-bit floating point (FP16), and 8-bit integer (INT8) as described in Table 6. During the training process, parameters and activation are represented in FP32 because FP32's high precision fares well for every training step that

**Table 6 Supported precision types of TensorRT.**

|  | Dynamic range | Min positive value | Note |
|---|---|---|---|
| FP32 | $-3.4 \times 10^{38}$ to $3.4 \times 10^{38}$ | $1.4 \times 10^{-45}$ | Training precision |
| FP16 | $-65{,}504$ to $65{,}504$ | $5.96 \times 10^{-8}$ | No calibration required |
| INT8 | $-128$ to $127$ | $1$ | Require calibration |

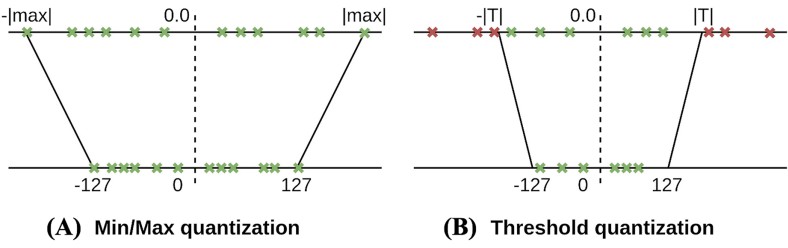

(A) Min/Max quantization          (B) Threshold quantization

**Figure 9 Comparison between the min/max quantization (A) and the threshold quantization (B).**

usually corrects the parameters by a small amount. However, in the inference phase, 16-bit or 8-bit representation of parameters and activation is preferred as lesser bit representation would need a lesser cycle for memory fetching and faster computation. Therefore, we need to perform a quantization process to re-encode the information to reduce the representation from 32 bits to 8 bits. The simplest method is the min-max quantization, where the min/max values of FP32 are mapped to min/max values of INT8, respectively. However, the min/max method leads to a significant accuracy loss. Therefore, instead of choosing the min/max value of FP32, they suggest to choose an appropriate threshold $T$ to map from $-T$ to $T$ in FP32 to $-128$ to $127$ in INT8. Figure 9 shows the difference between min/max quantization and threshold quantization. The intuition to choose the value $T$ which minimizes the loss of information between FP32 and INT8. The information loss can be considered as the "Relative Entropy" of two encodings and measured by Kullback-Leibler divergence (KL-divergence). To minimize the loss, they provide the calibration step to interpret a subset of the dataset to find the configuration.

After building the TensorRT engine for an optimized model, we execute the engine using Python APIs. The engine supports synchronous and asynchronous execution, profiling, enumeration, and querying of the bindings for inputs and outputs.

## Performance evaluation

We evaluate the performance of the models which are optimized with TensorRT on the Jetson board under various optimization profiles. Table 7 reports the performance of the eGAC3D model which uses DLA-34 architecture (*Yu et al., 2018*) as the backbone feature extractor. Additionally, we also develop a lightweight version of eGAC3D named eGAC3D-Lite that adopts ResNet-18 backbone (*He et al., 2016*). Table 8 summarises the performance of eGAC3D-Lite version. The comparison metrics include model size, inference time (not include pre/postprocessing time), the accuracy of 3D detection

**Table 7 Comparison of performance of eGAC3D (using DLA-34 backbone) with different optimization profiles.** We use bold to indicate the best result.

| Model | GPU | Mode | Size (MB) | Time (ms) | Accuracy $AP\|_{R40}^{3D}$ | | | Accuracy $AP\|_{R40}^{BEV}$ | | |
|---|---|---|---|---|---|---|---|---|---|---|
| | | | | | Easy | Mod | Hard | Easy | Mod | Hard |
| Pytorch | RTX 2080ti | FP32 | 82.1 | 34 | 21.02 | 16.34 | 14.08 | 31.29 | 22.91 | 19.92 |
| Pytorch | Xavier NX | FP32 | 82.1 | 491.6 | **21.02** | **16.34** | **14.08** | 31.29 | 22.91 | 19.92 |
| TensorRT-FP32 | Xavier NX | FP32 | 183.9 | 245 | 20.99 | **16.34** | 14.04 | **32.16** | 23.06 | **20.82** |
| TensorRT-FP16 | Xavier NX | FP16 | **55.4** | 161 | 20.82 | 16.29 | 13.97 | 31.99 | **23.70** | 20.74 |
| TensorRT-INT8 | Xavier NX | INT8 | 58 | **145** | 20.34 | 14.97 | 13.36 | 29.97 | 21.99 | 18.97 |

**Table 8 Comparison of performance of eGAC3D-Lite model (using ResNet-18 backbone) with different optimization profiles.** We use bold to indicate the best result.

| Model | GPU | Mode | Size (MB) | Time (ms) | Accuracy $AP\|_{R40}^{3D}$ | | | Accuracy $AP\|_{R40}^{BEV}$ | | |
|---|---|---|---|---|---|---|---|---|---|---|
| | | | | | Easy | Mod | Hard | Easy | Mod | Hard |
| Pytorch | RTX 2080ti | FP32 | 63.2 | 14 | 12.94 | 10.21 | 8.48 | 21.23 | 15.37 | 12.96 |
| Pytorch | Xavier NX | FP32 | 63.2 | 198 | 12.94 | **10.21** | **8.48** | 21.23 | 15.37 | **12.96** |
| TensorRT-FP32 | Xavier NX | FP32 | 134.9 | 88.8 | **13.00** | 10.20 | 8.46 | 21.33 | 15.38 | 12.90 |
| TensorRT-FP16 | Xavier NX | FP16 | **34** | 23.5 | 12.88 | 10.14 | 8.39 | **21.62** | **15.41** | 12.94 |
| TensorRT-INT8 | Xavier NX | INT8 | 34.1 | **18.4** | 11.17 | 8.30 | 6.84 | 18.47 | 13.20 | 11.70 |

$(AP\|_{R40}^{3D})$, and Bird-Eye-View $(AP\|_{R40}^{BEV})$. The first two rows show the original Pytorch model's performance when running on the mainframe computer with RTX-2080ti and the Jetson Xavier NX, respectively. The other three rows present the performance of the optimized models with TensorRT in three different precision modes.

It can be seen from Tables 7 and 8 that there is a severe gap in the latency when deploying the original Pytorch model on the Jetson board. The inference time escalates from 34 to 491.6 ms (for DLA-34 backbone) and from 14 to 209 ms (for ResNet-18 backbone). Although running with Jetson's GPU, the latency is still tremendous, and it is hard to apply it to real-time applications.

On the other hand, with the support of TensorRT, the inference time is significantly reduced. For the DLA-34 model, with the FP32 precision, the same precision when training the model, the TensorRT engine can run in just 245 ms, speeds up 2× times. If we quantize the precision to FP16 mode, the latency can reduce more than 3× times to 161 ms. For the ResNet-18 lightweight model, the efficiency of TensorRT optimization is more significant. The inference time can decrease up to 10× times, from 209 ms to 88.8 (FP32 mode) and 22.65 (FP16 mode), achieving real-time inference. INT8 mode gives the fastest results, but they are not noticeably improved when compared with FP16. The model's size can also be reduced with FP16 and INT8 mode. Figure 10 exhibits the throughput (in FPS) based on run-time results of the eGAC3D and eGAC3D-Lite models using Pytorch and TensorRT inference on Jetson Xavier NX.

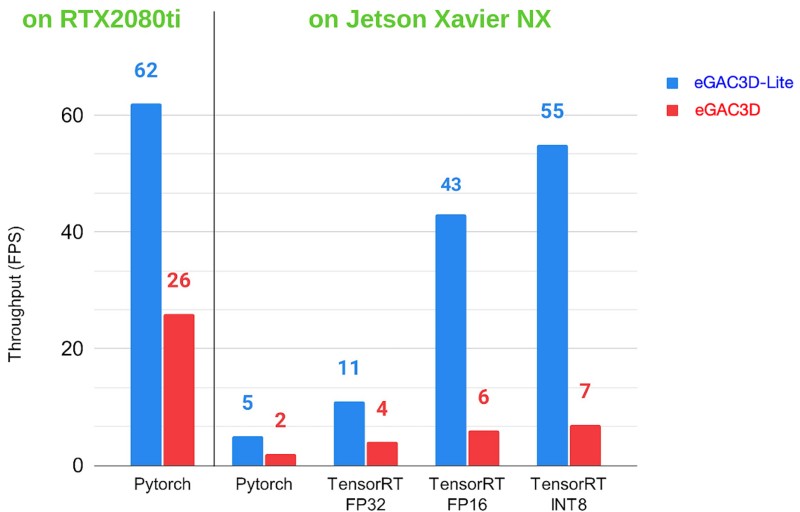

**Figure 10 Throughput (FPS) of our eGAC3D and eGAC3D-Lite models on Jetson Xavier NX with different optimization profiles.**

The FP32 and FP16 engines almost achieve the same performance as the original model in terms of accuracy. In INT8 precision mode, due to the information loss when quantizing the activation outputs of the model's layers, the model's accuracy is lower than the original model. As our model is used for detection task, the activation outputs of its layers, especially the final layers in the regression heads, are variant and sensitive. Therefore, quantizing from FP32 to INT8 leads to a significant gap in the model's final outputs then affects the accuracy.

## CONCLUSION

We presented an enhanced approach eGAC3D for monocular 3D object pose detection based on our previous work. By adopting a revised depth adaptive convolution with variant guidance formulated by Eq. (4), the detection accuracy has been improved. Additionally, by redesigning the architecture for jointly training the 3D detection and depth self-estimation to eliminate the external depth estimator, eGAC3D can significantly shorten inference time, which makes eGAC3D real-time inference. The experimental results have shown that eGAC3D outperforms many existing methods on the KITTI benchmark in terms of accuracy and time inference. Additionally, the experimental results in Table 5 also indicated that our proposed depth self-estimation quality can achieve better performance compared to other pre-trained depth models.

We were also the first to successfully deploy a monocular 3D detection framework on an embedded platform with the modest hardware resource, Jetson Javier NX embedded board. The experimental results have indicated that our proposed eGAC3D with TensorRT optimizer can significantly reduce both the model's size and latency to achieve real-time requirements while maintaining accuracy.

In future work, we will conduct more experiments on the embedded platform to finalize stable parameters for an optimization profile. Then, we will evaluate it on the data from the actual traffic scenarios and different datasets such as Waymo and CityScapes. Additionally,

the evolution of the deep neural network in computer vision also helps researchers improve backbone network architecture, which will be investigated to improve performance in 3D object detection tasks.

## ACKNOWLEDGEMENTS

We acknowledge Ho Chi Minh City University of Technology (HCMUT), VNU-HCM for supporting this study.

### Funding
This research is funded by the Vietnam National University Ho Chi Minh City (VNU-HCM) under grant number NCM2021-20-02. The funders had no role in study design, data collection and analysis, decision to publish, or preparation of the manuscript.

### Grant Disclosures
The following grant information was disclosed by the authors:
Vietnam National University Ho Chi Minh City (VNU-HCM): NCM2021-20-02.

### Competing Interests
The authors declare that they have no competing interests.

### Author Contributions
- Duc Tuan Ngo conceived and designed the experiments, performed the experiments, analyzed the data, performed the computation work, prepared figures and/or tables, and approved the final draft.
- Minh-Quan Viet Bui conceived and designed the experiments, performed the experiments, analyzed the data, performed the computation work, prepared figures and/or tables, and approved the final draft.
- Duc Dung Nguyen conceived and designed the experiments, performed the experiments, authored or reviewed drafts of the article, and approved the final draft.
- Hoang-Anh Pham conceived and designed the experiments, analyzed the data, authored or reviewed drafts of the article, and approved the final draft.

### Data Availability
The data are available at Zenodo: Hoang-Anh Pham. (2022). KITTI 3D Object Detection [Data set]. Zenodo. https://doi.org/10.5281/zenodo.7046095.

### Supplemental Information
Supplemental information for this article can be found online at http://dx.doi.org/10.7717/peerj-cs.1144#supplemental-information.

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
