# Peer review of "eGAC3D: enhancing depth adaptive convolution and depth estimation for monocular 3D object pose detection"

_PeerJ Computer Science, doi:10.7717/peerj-cs.1144_

## Round 0.1 · original submission · Major Revisions

The recommendations from the reviewers are not consensus. One accept, two minor, and one reject. As a result, the paper needs a further revision and clarification.

Reviewer 1 ·

Basic reporting

1. Your introduction needs more detail. I suggest that you improve the description at lines 64- 74 to provide more justification for your study. In addition, the description of the contribution at lines 77-95 need to be more concise. Finally, a description of the overall structure of the paper needs to be added.
2. The reasons and advantages of adding skip connections and using deformable convolution to DLA-34 of the backbone network are not explained. Please explain why you use this method.
3. The backbone network does not reflect the characteristics of the DLA-34 backbone network and is more like the structure of the hourglass network in figure 3. I suggest that you modify it to better express the characteristics of the network structure.
4. In the detection head part of the network structure, the construction rules for the number of convolution channels are not specified. Please provide detailed supplementary explanation.

Experimental design

no comments

Validity of the findings

no comments

Additional comments

The manuscript is clearly written in professional language. The research question is well defined, relevant and meaningful. If there is a weakness, it is the lack of detailed analysis of innovation points which should be improved upon before acceptance.

Reviewer 2 ·

Basic reporting

The paper is clear and professional, and the publication year of the references is relatively new, most of which are of high quality in recent years. In addition, the overall structure of the paper is in line with the standard, and the digital description is not cumbersome, so it meets the editing standard.

Experimental design

no comment

Validity of the findings

no comment

Additional comments

1、Your introduction needs more detail. I suggest that you improve the description at lines 185-187 to explain the advantages that you applied the deep layer aggregation backbone by adding more skip connections to the original DLA-34 network (Yu et al., 2018) and replacing the standard convolution layers in the upsampling nodes with the deformable convolution layers proposed in (Dai et al., 2017).
2、The variable f at line 204 is not reflected in formula (4), it is recommended to delete the description of variable f.
3、The acquisition of line 227 d* is not very clear, that is, the variable guidance function G is not very clear, you can consider adding a legend.
4、For the new head in the detection network mentioned at lines 232-245, its construction rules for the number of convolution channels are not specified. Design by experience or something else?

Reviewer 3 ·

Basic reporting

The language of this work is clear and concise in a good way for readers to understand the content and the connection with the previous work. The reference to related works are sufficient, and the authors even included a small analysis of different families of methods in Section 2. The tables and figures are well organized. The results reflected the main arguments of this paper.

Experimental design

The experimental design conforms to the norm used in the 3D perception world. The KITTI dataset is a good benchmark for such tasks' evaluation. In particular, I like the part where authors implemented the algorithm onto the Nvidia Jetson NX platform. Usually, 3D perception consumes lots of computational resources, and when arguing FPS, most works only test their results with fast performing serves or PCs. It is great that the authors of this work could use an embedded system for this test.

Validity of the findings

The data and experiments are sufficient to show the validity of the proposed methods.

Additional comments

This is the second work following the GAC3D. Looking forward to the next one.

Reviewer 4 ·

Basic reporting

As an improvement on the previous work there are some glaring issues in the writing.
Section Depth adaptive convolution layer has too many similarities over the previous work. A large portion of it was directly copied from GAC3D: improving monocular 3D object detection with ground-guide model and adaptive convolution. In this new paper, the description of the previous method should be summarized in a higher level instead of directly recycled.
Because of the recycled contents, the main focus of this paper is very unclear. More specifically, the improvements over the past methods are not highlighted, which made it really hard for people to understand the purpose of this new system.

Experimental design

The evaluation section is weak in this paper.
eGAC3D is only evaluated on KITTI, while many other recent methods have also been evaluated on newer datasets such as NuScenes.
When comparing performance, many state of the art methods are not included. For KITTI-3D test set evaluation on Car, many other methods can get an AP of above 30% or even 40%. But only the methods with lower 20% AP were included in the table. Also for the pedestrian and cyclists classes, there should be some explanations for getting an AP lower than 10%.

Validity of the findings

eGAC3D has a very limited amount of improvements over the old GAC3D according to the papers.
Both methods share the same overall structure while eGAC3D only improved mostly on additional detection heads and a revised confidence score.
It’s a bit too much to call this a separate study in my opinion. Also, as mentioned in the section above, eGAC3D’s performance is also not good enough to prove the validity of this method.

---

## Round 0.2 · accepted · Accept

The authors clarify the concerns raised by all of the reviewers, and appropriate revisions are done. The paper is ready to be accepted.

Reviewer 2 ·

Basic reporting

no comment

Experimental design

no comment

Validity of the findings

no comment

Additional comments

no comment